# Improvement in Dacryoendoscopic Visibility after Image Processing Using Comb-Removal and Image-Sharpening Algorithms

**DOI:** 10.3390/jcm11082073

**Published:** 2022-04-07

**Authors:** Sujin Hoshi, Kuniharu Tasaki, Kazushi Maruo, Yuta Ueno, Haruhiro Mori, Shohei Morikawa, Yuki Moriya, Shoko Takahashi, Takahiro Hiraoka, Tetsuro Oshika

**Affiliations:** 1Department of Ophthalmology, Faculty of Medicine, University of Tsukuba, 1-1-1 Tennoudai, Tsukuba 305-8575, Japan; k.tasaki1986@gmail.com (K.T.); yu_ueno71@yahoo.co.jp (Y.U.); moririn21@gmail.com (H.M.); s0711715@yahoo.co.jp (S.M.); yuki.moriya.429@gmail.com (Y.M.); shoko.takahashi.05@gmail.com (S.T.); thiraoka@md.tsukuba.ac.jp (T.H.); oshika@eye.ac (T.O.); 2Department of Biostatistics, Faculty of Medicine, University of Tsukuba, 1-1-1 Tennoudai, Tsukuba 305-8575, Japan; maruo@md.tsukuba.ac.jp

**Keywords:** dacryoendoscopy, lacrimal passage diseases, lacrimal sac, image enhancement, algorithms, image processing, lacrimal passage, dacryocystitis

## Abstract

Recently, a minimally invasive treatment for lacrimal passage diseases was developed using dacryoendoscopy. Good visibility of the lacrimal passage is important for examination and treatment. This study aimed to investigate whether image processing can improve the dacryoendoscopic visibility using comb-removal and image-sharpening algorithms. We processed 20 dacryoendoscopic images (original images) using comb-removal and image-sharpening algorithms. Overall, 40 images (20 original and 20 post-processing) were randomly presented to the evaluators, who scored each image on a 10-point scale. The scores of the original and post-processing images were compared statistically. Additionally, in vitro experiments were performed using a test chart to examine whether image processing could improve the dacryoendoscopic visibility in a turbid fluid. The visual score (estimate ± standard error) of the images significantly improved from 3.52 ± 0.26 (original images) to 5.77 ± 0.28 (post-processing images; *p* < 0.001, linear mixed-effects model). The in vitro experiments revealed that the contrast and resolution of images in the turbid fluid improved after image processing. Image processing with our comb-removal and image-sharpening algorithms improved dacryoendoscopic visibility. The techniques used in this study are applicable for real-time processing and can be easily introduced in clinical practice.

## 1. Introduction

Lacrimal passage obstruction may be congenital or acquired [1,2]. Its occurrence is concerning, because it can lead to lacrimal duct infections (such as dacryocystitis [3,4] and chronic conjunctivitis [5]) and significantly reduce the patient’s quality of life. Even in the absence of an obvious infection, epiphora due to lacrimal passage obstruction can reduce the vision quality and vision-related quality of life [6,7]. Surgical treatments include dacryocystorhinostomy [8] and the insertion of a silicone tube [9,10,11] or the Jones tube [12,13]; however, the exact location of the obstruction must be identified for appropriate treatment selection [14,15,16,17].

In addition to classical diagnostic methods (such as lacrimal irrigation, blind probing, and dacryocystography), dacryoendoscopy has been recognized as an effective diagnostic tool [18,19]. In dacryoendoscopy, the lacrimal passage is directly visualized using a dacryoendoscope, which is inserted into the lacrimal passage through the lacrimal punctum. This is useful for the diagnosis and treatment of various lacrimal passage diseases, such as congenital [1,20], acquired [10,21], and secondary [22,23] lacrimal duct obstruction; canaliculitis [24]; tissue granulation [25]; and tumors [26]. Fiberscope-type dacryoendoscopes were introduced in the 1990s and had a pixel count of 1800–3000 [27,28,29]. Subsequently, 6000-pixel dacryoendoscopes were introduced in 2002 [30], while 10,000-pixel and 15,000-pixel dacryoendoscopes were introduced in 2009 [16].

Better visualization of the lacrimal canal with a dacryoendoscope improves the analysis of the lacrimal passage conditions and allows the precise identification of the obstructing lesions, both of which are key to a successful surgery [16,17]. The increase in the number of pixels has led to an improvement in the diagnostic performance of dacryoendoscopes; however, the image quality still requires improvement for an enhanced therapeutic performance [18].

Two main factors limit the quality of fiberscope-type dacryoendoscopic images. One is the limited number of individual fiberlets, which define the number of pixels. The other is the small amount of perfusion fluid that is needed to sufficiently flush away the blood and pus clouding the lacrimal duct. The number of fiberlets and amount of perfusion fluid are determined by the diameter of the dacryoendoscope probe. Increasing the probe diameter can improve the image quality by placing more fiberlets and providing a larger perfusion channel. However, the probe diameters of the current mainstream dacryoendoscopes range from 0.7 mm to 1.1 mm; it is difficult to increase the diameter further due to the small size of the punctum.

We proposed that the imaging quality of fiberscope-type dacryoendoscopes can be improved by image processing. Images obtained from fiberscope-type endoscopes contain black fine mesh noise (a comb artifact), which is caused by the opaque cladding between the fiberlets; this artifact adversely affects the visibility [31,32]. This problem is especially critical in case of dacryoendoscopes, which have a limited number of pixels as compared to the endoscopes used for other organs.

This study aimed to remove the aforementioned comb artifact from dacryoendoscopic images and reduce visibility loss due to blood- or pus-associated opacity in the lacrimal duct by using an algorithm capable of real-time image processing. Thus, we visually evaluated and compared the dacryoendoscopic visibility before and after image processing to assess the usefulness of this algorithm.

## 2. Materials and Methods

This single-center, observational study was approved by the Ethics Committee of the Faculty of Medicine, University of Tsukuba (R03-101). It adheres to the tenets of the Declaration of Helsinki.

We used images that had already been acquired previously and did not contain any personal information; therefore, patient consent was obtained in an opt-out format.

### 2.1. Image Samples

The images were arbitrarily extracted from the dacryoendoscopic images acquired and recorded within the scope of standard medical care at the University of Tsukuba Hospital between April 2019 and March 2021.

The dacryoendoscopic images were acquired by five surgeons (T.H., S.H., K.T., H.M. and S.T.). A close-up dacryoendoscope with a focal length of 1.5 mm (LAC-06NZ-HS; MACHIDA Endoscope Co., Ltd., Chiba, Japan) and a high-definition camera (MVH-2010A; MACHIDA Endoscope Co., Ltd., Chiba, Japan) were used for the imaging; this setting produced 16,000-pixel images.

Twenty original images were prepared by arbitrarily extracting five sample images of the canaliculus, lacrimal sac, nasolacrimal duct, and intra-canal materials; the exposure time for each was 10–20 s. In case of the canaliculus, lacrimal sac, and nasolacrimal ducts, images were selected after confirming that the luminal structure was visible and that there was very little or no bleeding- or pus-associated opacity. Images of the intra-canal materials included images of a foreign body straying into the lacrimal canal, those of a silicone tube implanted in the lacrimal canal, and those taken during sheath-guided endoscopic probing [33]. A total of 40 images were subjected to visual evaluation; these comprised the 20 original images and their post-processing versions (*n* = 20).

### 2.2. Image Processing

The dacryoendoscopic images were processed using comb-removal (WipeFiber^®^; Logic & Design Inc., Tokyo, Japan, and MACHIDA Endoscope Co., Ltd., Chiba, Japan) and image-sharpening (Medical Image Enhancer: MIEr^®^; Logic & Design Inc., Tokyo, Japan) algorithms. The image captured by a fiberscope is a collection of small circular images that are captured in turn by the fiberlets, and the fiberlets’ boundaries are visible as a comb-structured artifact. This comb-structured artifact is considered the main cause of a reduced visibility during dacryoendoscopy. Comb removal was originally developed as a technique for improving the imaging quality of endoscopes for other organs, such as gastrointestinal endoscopes; however, similar techniques can be applied to dacryoendoscopes. The noise caused by the comb structure appears as a high-frequency component in the spatial frequency domain and can be removed by a low-pass filter. In this study, we used a Gaussian filter, which is a high-performance and an easy-to-implement low-pass filter. Subsequently, the image was sharpened by applying contour enhancement only in the specific frequency region according to the fiberlet diameter, while simultaneously suppressing the restoration of the removed comb structure. The image sharpening process was also used to improve the visibility in low-contrast areas.

The algorithm used in this study could perform real-time image processing; however, real-time processing was not implemented in this study.

### 2.3. Image Visibility Evaluation

Visual evaluation was performed for all images, and the visibility was scored on a 10-point scale by the evaluator. Three evaluators were presented with the 20 original and the 20 processed images in a random order, one at a time. The images were presented on a 13.3-inch monitor (2560 × 1600-pixel resolution), and the images (>15 cm in diameter) were sequentially displayed on a black background.

The results were analyzed statistically, and comparisons were performed between the images before and after image processing. These comparisons were made separately for all images, including those of the canaliculus, lacrimal sacs, nasolacrimal ducts, and intra-canal materials. A linear mixed-effects model was applied, which included the visibility score as the outcome variable; the processing, image type, and their interaction as the fixed effects; and the image ID and evaluator as the random effects. Subsequently, inferences on the least square mean differences between before and after processing were drawn for all data and each image type. The statistical significance was set at *p* < 0.05. All statistical analyses were performed using the SAS software (version 9.4; SAS Institute, Cary, NC, USA) and SPSS (version 23.0; IBM, Armonk, New York, NY, USA).

### 2.4. In Vitro Experiments

Image processing comprised comb removal and image sharpening. In vitro experiments were performed to assess the effects of image processing on the deterioration of dacryoendoscopic image quality due to opacity caused by the presence of blood and pus in the lacrimal duct lumen.

We prepared diluted milk saline (DMS), at concentrations of 1%, 2%, and 3%, to simulate turbidity in the lacrimal pathway. We then prepared 50% diluted blood (defibrinated sheep blood, AS ONE Corporation, Osaka, Japan) to simulate hemorrhage in the lacrimal pathway. Saline was used as a control for the test solution. We evaluated the image resolution (line pairs/mm) by imaging a test chart (3M550, Pearl Optical Industry, Tokyo, Japan), as described previously [17].

The test solution was placed on the test chart under a cover glass, and the dacryoendoscope was fixed such that the tip of its probe was positioned 1.5 mm above the test chart. The images were acquired after confirming that the probe’s tip was in contact with the test solution. The power of resolution in each condition from the test chart images were evaluated by two examiners (S.H. and K.T.).

## 3. Results

Figure 1 shows the original and post-processing images from a representative case; further details are available in Appendix A.

The comb-structured artifact was observed in the original images, which were uneven and difficult to observe (Figure 1a,c,e,g; Appendix A). However, this artifact was removed by image processing, and the visibility was improved in the post-processing images (Figure 1b,d,f,h; Appendix A).

For example, Figure 1c shows the original image of the lacrimal sac (Figure 1c; Appendix A); the blood vessels in the lacrimal sac mucosa are more highlighted in the corresponding processed image (Figure 1d; Appendix A).

A comparison of the nasolacrimal duct images revealed that the luminal structures that were obscured in the original image (Figure 1e; Appendix A) were visible in the processed image (Figure 1f; Appendix A).

Figure 2 shows a graph comparing the visual scores of all images. The visual score (estimate ± standard error) significantly improved from 3.52 ± 0.26 (original images) to 5.77 ± 0.28 (post-processing images; *p* < 0.001, linear mixed-effects model). Similarly, the visual scores of the original and post-processing images were 2.73 ± 0.45 and 5.07 ± 0.60 for the canaliculus, 4.07 ± 0.47 and 6.33 ± 0.48 for the lacrimal sac, 4.33 ± 0.51 and 6.47 ± 0.47 for the nasolacrimal duct, and 2.93 ± 0.46 and 5.20 ± 0.53 for the intra-canal materials, respectively. After image processing, all scores improved significantly (*p* < 0.001, linear mixed-effects model).

The apparent visibility-improvement effect of image processing was confirmed in the in vitro experiment (Figure 3).

Under saline, the original image (Figure 3c) showed prominent comb artifacts; however, in the processed image (Figure 3d), the comb artifacts were eliminated, and the visibility was improved. The image resolution of the original image was 25 line pairs/mm (Figure 3c), whereas that of the processed image was 28 line pairs/mm (Figure 3d).

The images observed with 1%, 2%, and 3% DMS showed an improved contrast and resolution after image processing. The resolutions of the original and processed images were as follows: 1% DMS, 10 line pairs/mm (Figure 3e) vs. 22 line pairs/mm (Figure 3f); 2% DMS, 10 line pairs/mm (Figure 3g) vs. 16 line pairs/mm (Figure 3h); and 3% DMS, not evaluable (Figure 3i) vs. 3 line pairs/mm (Figure 3j).

The image observed with 50% blood showed an improved visibility after image processing. However, the resolution barely differed between the original and processed images (18 line pairs/mm (Figure 3k) vs. 18 line pairs/mm (Figure 3l)).

## 4. Discussion

The image-processing algorithm used in this study improved the dacryoendoscopic visibility for every region of the lacrimal duct lumen. The dacryoendoscope has a small diameter and can deliver a limited amount of light; thus, light cannot reach regions as deep as the nasolacrimal duct. It is often difficult to observe the details in a deep duct (Figure 1e). However, the contrast between components can be improved by image processing. As shown in Figure 1f (post-processing image), it was possible to visualize the details in the luminal structure that were not visible in Figure 1e (original image). Accurate identification of the site of lacrimal passage obstruction is important for its successful treatment with lacrimal endoscopy [16]. Image processing allows a clear visualization of the dimple sign, which indicates an obstruction of the lacrimal duct; this in turn is expected to prevent the creation of false passages and enable an accurate recanalization of the lacrimal duct.

Recently, various microendoscopic transcanalicular therapies have been developed for the management of lacrimal passage obstruction; these include transcanalicular endoscopic dacryoplasty, dacryoendoscopic-assisted nasolacrimal duct intubation [9,10,11], electrocautery-based techniques [34], and laser or microdrill dacryoplasty [35]. To perform these techniques adequately, it is important to improve the visibility of the dacryoendoscope. Enhancement of the luminal structure images by image processing is expected to improve dacryoendoscopic treatment outcomes for lacrimal passage diseases.

Dacryoendoscopy is attracting increasing attention for performing detailed observations of the lacrimal duct lumen mucosa. Mimura et al., demonstrated that staining of the lacrimal canal mucosa with indigo carmine during dacryoendoscopic examination enabled the assessment of fibrosis and inflammation [36]. In the present study, the blood vessels in the lacrimal sac images were more clearly visible after image processing (Figure 1c,d). Thus, this technique would allow a deeper understanding of lacrimal duct pathologies by enabling a more detailed observation of the lacrimal duct mucosa.

The reduced visibility due to blood- or pus-associated clouding during dacryoendoscopic observation was reportedly improved by using the dacryoendoscope under air perfusion [37]. However, performing lacrimal endoscopy under air perfusion requires a special setting, which is not available commercially and is difficult to establish in general practice. Conversely, the algorithms used in this study can be easily introduced into clinical practice. They can be integrated into an existing video system to perform real-time image processing. In addition, the in vitro experiments performed in our study demonstrated a high image-sharpening effect of these algorithms on the blood- and saline-associated opacity; this is expected to be useful in actual clinical practice.

This study had some limitations. First, the study design did not facilitate the direct evaluation of the improvements in the diagnostic accuracy and treatment outcomes following image processing. Further studies will be needed to clarify whether the proposed algorithms contribute to improved clinical outcomes. Second, other expected benefits, such as a shortened treatment time and reduced surgeon stress, could not be determined due to the study’s retrospective nature. Therefore, prospective studies on patients are desirable in order to investigate this. Furthermore, to ensure that this modality benefits the patients, it is important that the image processing device becomes widely available at low additional costs.

## 5. Conclusions

In conclusion, this study demonstrated that image-processing and image-sharpening algorithms for comb artifact removal could improve the dacryoendoscopic visibility and thus potentially improve the diagnostic accuracy and treatment outcomes in patients with lacrimal passage obstruction.

## Figures and Tables

**Figure 1 jcm-11-02073-f001:**
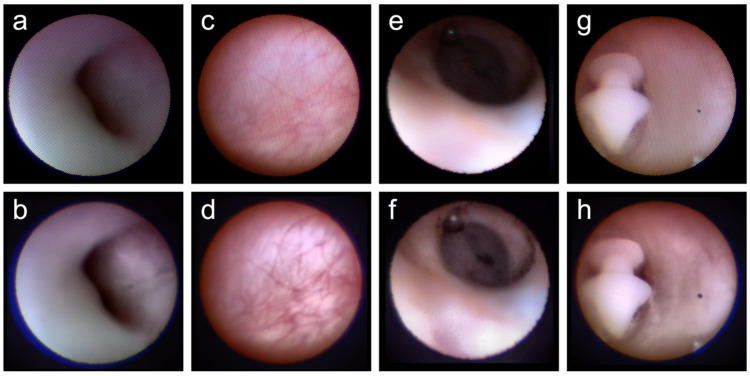
Original and post-processing representative images of the canaliculus (**a**,**b**), lacrimal sacs (**c**,**d**), nasolacrimal ducts (**e**,**f**), and intra-canal materials (**g**,**h**). The comb-structured artifact was observed in the original images, which were uneven and difficult to observe (**a**,**c**,**e**,**g**). However, in the post-processing images, this artifact was removed by image processing, and the visibility was improved (**b**,**d**,**f**,**h**). The blood vessels in the lacrimal sac mucosa were more highlighted in the processed image than in the original image (**c**,**d**). On comparing the nasolacrimal duct images, the luminal structures that were obscured in the original image (**e**) were visible in the processed image (**f**).

**Figure 2 jcm-11-02073-f002:**
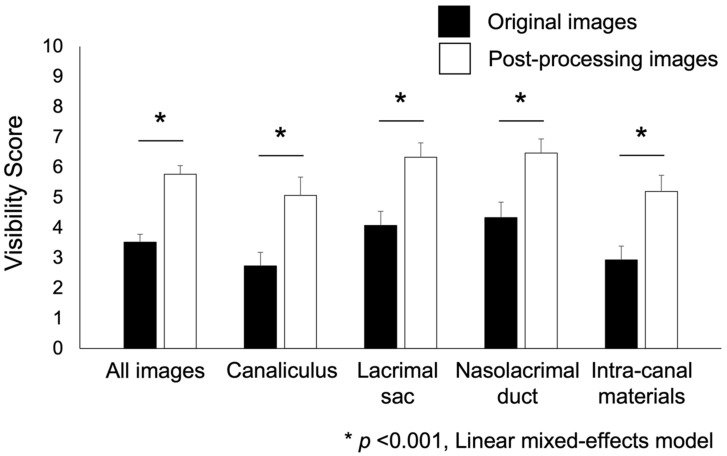
A graph comparing the visual scores of all images before and after image processing, including those of the canaliculus, lacrimal sacs, nasolacrimal ducts, and intra-canal materials.

**Figure 3 jcm-11-02073-f003:**
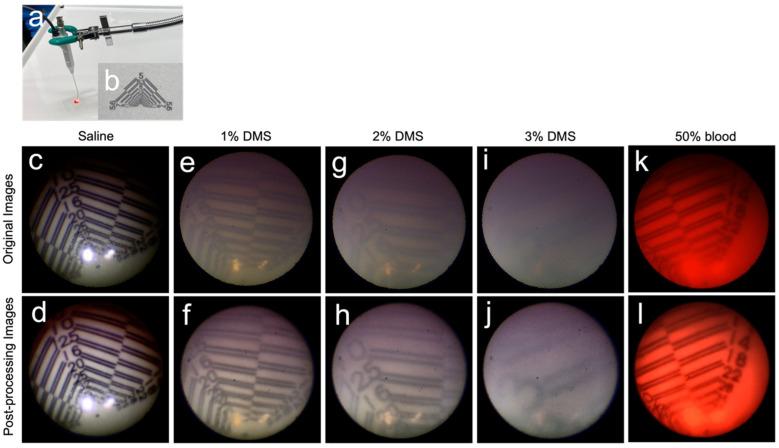
Settings and images of the in vitro experiment conducted to assess the effects of image processing on dacryoendoscopic images, whose quality deteriorated due to opacity caused by blood and pus in the lacrimal duct lumen. The test solution was placed on the test chart with a cover glass, and the endoscope was positioned such that the tip of its probe rested 1.5 mm above the test chart (**a**,**b**). Under saline, the original images (**c**) showed prominent comb artifacts. However, in the processed images (**d**), these comb artifacts were eliminated, and the visibility was improved. The image resolution of the original image was 25 line pairs/mm (**c**), whereas that of the processed image was 28 line pairs/mm (**d**). Images observed with 1%, 2%, and 3% diluted milk saline (DMS) showed an improved contrast and resolution after image processing. The resolutions of the original and post-processing images were as follows: (1) 1% DMS, 10 line pairs/mm (**e**) vs. 22 line pairs/mm (**f**); (2) 2% DMS: 10 line pairs/mm (**g**) vs. 16 line pairs/mm (**h**); and (3) 3% DMS: not evaluable (**i**) vs. 3 line pairs/mm (**j**). The resolution of the original and processed images in 50% blood was 18 line pairs/mm (**k**,**l***)*.

## Data Availability

The data presented in this study are available on request from the corresponding author.

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
