# Peer review of "Improvement in Dacryoendoscopic Visibility after Image Processing Using Comb-Removal and Image-Sharpening Algorithms"

_jcm, 2022, doi:10.3390/jcm11082073_

Round 1

Reviewer 1 Report

The introduction is too large for the purpose of the study, it should be shortened.

By applying image processing, there is an improvement in the identification of structures according to the analysis of the authors. However, this requires a technology not available in most hospitals, so it is not very reproducible, in addition to the fact that it would make services more expensive.

It should be verified that the proposed technology represents an improvement in the diagnosis and treatment of patients compared to the current method.

Author Response

Response to Reviewer 1 Comments

Point 1: The introduction is too large for the purpose of the study, it should be shortened.

Response 1: Thank you for your comment. In accordance with your suggestion, we have revised the Introduction by removing statements that were unnecessary for explaining the purpose of the study. However, please take it into account that one of the reviewers has asked us to enhance the Introduction section by adding more details on Dacryoendoscopy and its applications.

Point 2: By applying image processing, there is an improvement in the identification of structures according to the analysis of the authors. However, this requires a technology not available in most hospitals, so it is not very reproducible, in addition to the fact that it would make services more expensive.

Response 2: Thank you for your insightful comment. As you have noted, this technology is not available in most hospitals; in fact, it is only available in Japan in select hospitals. To ensure that the technology benefits the patients, the image processing device must become widely available in the region at a low additional cost. This has been addressed as a study limitation in the Discussion.

Point 3: It should be verified that the proposed technology represents an improvement in the diagnosis and treatment of patients compared to the current method.

Response 3: We thank you for this important comment. We agree with you it is necessary to confirm that compared to the currently available methods, the proposed algorithms provide more improved outcomes. However, we could not do so due to the study’s retrospective design. We believe that this the most critical limitation of our study; it has been addressed in the Discussion section accordingly. Please see lines 268–271:

“This study had some limitations. First, the study design did not facilitate the direct evaluation of the improvements in the diagnostic accuracy and treatment outcomes following image processing. Further studies will be needed to clarify whether the proposed algorithms contribute to improved clinical outcomes.”

Reviewer 2 Report

This important paper describes the utilization of image processing to reduce artifacts and improve the visibility of dacryoendoscopic images. The paper is well designed and written. A few minor comments: 

  1. Comb artifact & comb removal - these require further explanation and elaboration. Please present the comb artifact as early as in the introduction section. Also, please describe the comb removal - including for what purpose was this originally designed. 
  2. Figure 3: please explain the differences between c-k – are these different dilutions? 

Author Response

Response to Reviewer 2 Comments

Point 1: Comb artifact & comb removal - these require further explanation and elaboration. Please present the comb artifact as early as in the introduction section. Also, please describe the comb removal - including for what purpose was this originally designed.

Response 1: Thank you for your comments. We have added an explanation of the comb artifact in the Introduction section (lines 75–77), and have also described it in further detail in the Materials and Methods section (lines 117–120). Comb removal was originally developed to improve the imaging quality of endoscopes for other organs, such as gastrointestinal endoscopes; however, similar techniques can be applied to dacryoendoscopes. We have presented the details of comb removal in the Materials and Methods section (lines 120–123).

Point 2: Figure 3: please explain the differences between c-k – are these different dilutions?

Response 2: Thank you for your question. Yes, Figures 3c–3k represent images taken under different solutions at different dilutions. Original and post-processing images were taken under saline (c and d, respectively), 1% diluted milk saline (DMS; e and f, respectively), 2% DMS (g and h, respectively), and 3% DMS (i and j, respectively), and 50% blood (k and l, respectively). To ensure that this is easier for the readers to understand, we have added labels for each image in Figure 3.

Reviewer 3 Report

Authors presented a research paper on "Improvement in Dacryoendoscopic Visibility with Image Processing using Comb Removal and Image Sharpening Algorithms" Methodology and experimentation is found to be good. Clinical translation of this research will be helpful for patients. Manuscript can be accepted with the following corrections. 

  1. Please elaborate "Introduction" section by providing more information and literature regarding application of Dacryoendoscopy similar applicaiton.
  2. Request to check for few grammatical errors in the manuscript. 

Author Response

Response to Reviewer 3 Comments

Point 1: Please elaborate "Introduction" section by providing more information and literature regarding application of Dacryoendoscopy similar applicaiton.

Response 1: Thank you for this comment. In accordance with your suggestion, we have cited additional literature on the application of dacryoendoscopy in the Introduction section (please see lines 43–55):

“In dacryoendoscopy, the lacrimal passage is directly visualized using a dacryoendoscope, which is inserted into the lacrimal passage through the lacrimal punctum. This is useful for the diagnosis and treatment of various lacrimal passage diseases, such as congenital [20 ,1], acquired [21, 10], and secondary [22, 23] lacrimal duct obstruction; canaliculitis [24]; tissue granulation [25]; and tumors [26].”

However, please take it into account that one of the reviewers has asked us to reduce the Introduction length; therefore, we have remained brief in our explanations.

Point 2: Request to check for few grammatical errors in the manuscript.

Response 2: Thank you for flagging this with us. The revised manuscript was edited for English language by Editage.

Reviewer 4 Report

Overall, the research showed improved image clarity after the sharpening algorithm. It would be great if the authors could explain in details how the technology can be used to improve the diagnosis of lacrimal passage disease. Otherwise, the paper will lose its readability and correlation with only image processing. 

Author Response

Response to Reviewer 4 Comments

Point 1: Overall, the research showed improved image clarity after the sharpening algorithm. It would be great if the authors could explain in details how the technology can be used to improve the diagnosis of lacrimal passage disease. Otherwise, the paper will lose its readability and correlation with only image processing. 

Response 1: We thank you for your important comment. We have explained how the proposed technology can contribute to the appropriate diagnosis and treatment of lacrimal passage diseases. Please see lines 224–227:

“Image processing allows a clear visualization of the dimple sign, which indicates an obstruction of the lacrimal duct; this in-turn is expected to prevent the creation of false passages and enable an accurate recanalization of the lacrimal duct.”